# SARS-CoV-2 Affects Both Humans and Animals: What Is the Potential Transmission Risk? A Literature Review

**DOI:** 10.3390/microorganisms11020514

**Published:** 2023-02-17

**Authors:** Antonio Santaniello, Giuseppe Perruolo, Serena Cristiano, Ayewa Lawoe Agognon, Serena Cabaro, Alessia Amato, Ludovico Dipineto, Luca Borrelli, Pietro Formisano, Alessandro Fioretti, Francesco Oriente

**Affiliations:** 1Department of Veterinary Medicine and Animal Production, Federico II University of Naples, 80134 Naples, Italy; 2Department of Translational Medical Sciences, Federico II University of Naples, 80131 Naples, Italy

**Keywords:** SARS-CoV-2, individuals, companion animals, animal-assisted interventions, bidirectional transmission risk

## Abstract

In March 2020, the World Health Organization Department declared the coronavirus (COVID-19) outbreak a global pandemic, as a consequence of its rapid spread on all continents. The COVID-19 pandemic has been not only a health emergency but also a serious general problem as fear of contagion and severe restrictions put economic and social activity on hold in many countries. Considering the close link between human and animal health, COVID-19 might infect wild and companion animals, and spawn dangerous viral mutants that could jump back and pose an ulterior threat to us. The purpose of this review is to provide an overview of the pandemic, with a particular focus on the clinical manifestations in humans and animals, the different diagnosis methods, the potential transmission risks, and their potential direct impact on the human–animal relationship.

## 1. Introduction

Zoonoses are infections that naturally spread from animals to humans and vice versa [1].

After the last ice age (10–12,000 years ago), when nomadic groups of hunter-gatherers became permanent farmers, the domestication of plants and animals promoted the passage of microorganisms to humans. Consequently, the human immune system found itself facing these pathogenic parasites for the first time, unlike the animals that for millennia had learned to live with them [2,3].

Globalization, population density, traveling, climate change, work, and relationships have played a significant role in the emergence and spread of zoonoses. These diseases can be transmitted in many ways, including animal bites and insect stings, petting, or otherwise coming into contact with sick animals, as well as consuming undercooked meat, unpasteurized milk, or contaminated water.

The type of pathogens that can be transmitted from animals to humans include bacteria, parasites, fungi, and viruses. According to WHO (World Health Organization), there are more than 200 known zoonotic diseases and 6 out of 10 cases of infectious diseases reported each year have a zoonotic origin. Moreover, zoonoses are estimated to be responsible for 2.5 billion cases of disease and 2.7 million deaths worldwide each year [4]. Indeed, although some zoonotic diseases are relatively benign, such as encephalitozoon cuniculi infections [5] or dermatophytosis [6], several others are quite harmful.

Among the worst zoonoses, there is that of the black death (also known as the bubonic plague), which took the lives of 50 million people in the 14th century. The etiology of this zoonosis is the bacillus Yersinia pestis which is transmitted from rodents to humans via flea bites. The expression “Black Death” is derived from the observation of dark and livid spots of hemorrhagic origin that manifested themselves on the skin and mucous membranes of the sick and also from the blackened tissue due to gangrene [7]. Very few cases of bubonic plague still occur throughout the world. This disease can be treated and cured with antibiotics, including ciprofloxacin, levofloxacin, moxifloxacin, gentamicin, and doxycycline. Vaccines are available for use in laboratory staff working on the disease. However, the effectiveness or tolerability of any plague vaccines are still debated [8].

Another millenary zoonosis is rabies which is caused by lyssaviruses. This disease can spread to people through wild or domestic animal bites or scratches. Weakness, fever, headache, cerebral dysfunction, anxiety, confusion, and agitation are the most common symptoms of rabies. As the disease progresses, paralysis and coma can follow until death. Rabies occurs in more than 150 countries and territories. According to an estimation by WHO, almost 55,000 people die because of rabies every year. Once a rabies infection has been established, there is no effective treatment. Fortunately, a vaccine is available for people at risk of being infected. Thus, the post-exposure treatment consists, first of all, of cleaning and disinfecting the wound, and then giving a person an injection of rabies immune globulin and another injection of rabies vaccine as soon as possible after the bite [9].

Human prion disease is a rare, fatal neurodegenerative disease, whereby major subtypes of which include bovine spongiform encephalopathy (BSE), fatal familial insomnia (FFI), and Gerstmann–Sträussler–Scheinker disease (GSS). Prion disease is the cause of roughly 1 in 6000 deaths, with an incidence of one to two cases per million population per year [10,11]. In particular, BSE is a recently discovered zoonotic disease that affects adult cattle. Commonly known as “mad cow disease” because it was detected in cattle in the U.K. in 1986, BSE attacks the brain and central nervous system of the animal and eventually causes death. In 1996, a new variant of Creutzfeldt–Jakob disease (CJD) was identified in humans and was presumed to be caused by the consumption of contaminated meat and other food products derived from affected cattle. CJD can induce behavioral and personality changes, confusion and memory problems, depression, insomnia, lack of coordination, and vision problems, symptoms that may worsen, leading to death. Actually, there is no treatment or vaccine to prevent BSE. Control measures are the banning of meat and bone meals in cattle feeds, active and passive surveillance, and the culling of sick animals.

Acquired immunodeficiency syndrome (AIDS) is a chronic, potentially life-threatening condition caused by the human immunodeficiency virus (HIV). Interestingly, based on findings demonstrating that HIV developed as a result of multiple cross-species transmissions from simian immunodeficiency virus (SIV), AIDS has been reported to be the most important zoonosis in our recent history, killing more than 25 million people [1,12,13]. HIV has probably “jumped” to humans from a West African subspecies of chimpanzee (*Pan troglodytes troglodytes*). It causes a progressive weakening of the immune system, making it vulnerable to pathogenic microorganisms and tumors. The first recorded cases were in 1981 and the virus has hit all countries, particularly those of the Third World. Currently, the treatment of HIV infection consists of a drug combination that blocks the replication of the virus, since there is no vaccine [14,15,16,17].

A novel zoonotic disease was reported in December 2019 in Wuhan (China). Patients were presumably exposed to a new coronavirus, designated as SARS-CoV-2, around the Huanan Seafood Wholesale Market in China, where vendors of live wild animals congregated and where virus-positive environmental samples were concentrated [18]. This virus probably crossed into humans from an animal species, most likely a bat, spreading rapidly across the world and causing the coronavirus disease 2019 (COVID-19) [19]. This infectious illness was responsible for a global pandemic that caused deaths and economic despair. The latest WHO data (February 2023) confirmed 755,703,002 cases worldwide since the beginning of this zoonosis and 6,836,825 deaths, making this pandemic one of the deadliest in human history.

The purpose of this review is to provide a comprehensive analysis of the scientific literature surrounding the presence of SARS-CoV-2 in animals and humans, evaluating the potential transmission risk.

## 2. Methods

For our review, a recent scientific literature search was performed until 30 November 2022, referring to the National Library of Medicine “PubMed.gov” for the words “SARS-CoV-2—virus” or “COVID-19” and “animals”, “dog”, “cat”, or “human”. The authors selected articles that described the epidemiology, the potential susceptible animal species (domestic or wild), and diagnostic techniques related to SARS-CoV-2 virus. The titles of all found articles were screened for relevance with respect to the topic, and then appropriate titles were assessed and selected based on their abstracts. Additional studies were found using the references of the selected papers. Only original papers in English were included.

## 3. Coronavirus Disease 2019 (COVID-19)

### 3.1. SARS-CoV-2 Virus

Coronaviruses (CoVs) are a large family of single-stranded RNA viruses [20]. These viruses have been known for several years. Indeed, the first coronavirus of human interest, the B814 virus, was isolated from the mucus of a patient suffering from a common cold and described in 1965 [21]. CoVs belong to the Coronaviridae family of the Nidovirales order. The subfamily Coronavirinae can be further divided into four genera: Alpha, Beta, Gamma, and Delta CoVs. Gamma and Delta coronaviruses infect a wide range of animal species, particularly avian ones. The Alpha and Beta coronaviruses mainly infect mammals, including humans, and typically cause transient respiratory or gastrointestinal illness. In recent years, however, it has emerged that CoVs can cause more severe and potentially fatal diseases of the respiratory system, such as MERS-CoV (Middle East respiratory syndrome), SARS-CoV (severe acute respiratory syndrome) and SARS-CoV-2 (severe acute respiratory syndrome 2) [20]. Among these viruses, SARS-CoV-2 was identified as the source of a pneumonia outbreak in Wuhan, China, in late 2019 [22]. The COVID-19 incubation period is about 5–6 days and, in some cases, is more than 14 days; during this time, virus transmission can occur via direct (deposited on persons) or indirect (deposited on objects) contact and airborne (droplets and aerosols) routes [23,24]. Following the sequencing of the SARS-CoV-2 genome, a high homology with the genome of some CoVs that infect bats was highlighted, leading to the hypothesis of a possible transmission of the virus from bats to an intermediate host, such as the human being. When observed under the electron microscope, SARS-CoV-2 appears round or oval in shape with a diameter of about 60–100 nm, and it has a crown-like appearance which gives it the Latin name corona, meaning crown or halo. SARS-CoV-2 is a non-segmented positive-stranded RNA virus whose genome is around 30 kb [25]. The coronavirus genome encodes 25 nonstructural and 4 structural proteins. The nonstructural and accessory proteins facilitate viral replication and transcription, release of virus particles, and carrying to the host cells. The structural proteins, which include the spike (S), membrane (M), envelope (E), and nucleocapsid (N) proteins, are involved in morphogenesis, viral assembly, host infection, and membrane fusion (Figure 1). In more detail, the M protein helps the virus to evade host antiviral innate immunity [26], crosses the envelope, and interacts with the RNA-protein complex in the virion. The N protein is engaged in the binding and bundle of the RNA genome and together with the M protein cooperates in the shaping of viral particles [27]. The S protein mediates the entry of the virus into the host cell, drives the virus infection efficacy of human cells and is sought after as a major target for antiviral drugs [25]. When in contact with the target cell, SARS-CoV-2 binds to the ACE2 (Angiotensin-converting-enzyme 2) receptor through the S1 subunit of the protein S. Via the S2 subunit, SARS-CoV-2 fuses together with the target cell membrane and releases its genome into the cell. Once into the host cell’s cytoplasm, a transcription complex synthesizes the double-stranded RNA (dsRNA) genome from the genomic ssRNA (+). The dsRNA genome is transcribed and replicated to create viral mRNAs and new ssRNA (+) genomes [28]. Finally, the produced structural proteins assemble to form new virions which can be secreted by the infected cell.

### 3.2. Infections in Animals

Coronavirus infections of veterinary interest have been known for almost a century [29,30]. CoVs are especially known for their genetic plasticity, which allows them to generate strains with different biological properties; this mechanism allows them to have a wide range of hosts [31]. In the current scientific literature, there are several data which suggest the susceptibility of companion and wild animals, including the characteristics of the host cell receptors to which the virus binds and the demonstrations of experimental and acquired infections [32]. The last update published by FAO dating back to December 2022 mentions about 30 animal species naturally infected (RNA detection) by SARS-CoV-2. SARS-CoV-2 infection in animals is usually asymptomatic but can occasionally cause anything from symptoms ranging from mild respiratory and gastrointestinal symptoms to pneumonia and death [33].

Among the factors that influence the inter-specific contagiousness of SARS-CoV-2 are the polymorphisms of the genes that code for the receptors of the animal cells to which the virus binds. Several proteins fundamental for viral susceptibility have been categorized. As reported above, angiotensin-converting enzyme 2 (ACE2) is a transmembrane protein that acts as a functional receptor of the spike protein (S) for SARS-CoV-2 virus entry into the cell [34,35]. However, there are other proteases involved in this mechanism. The role of the Serine Type2 transmembrane protease (TMPRSS2) is noteworthy. It has been recognized that both ACE2 and TMPRSS2 are required for the entry of SARS-CoV-2 into host cells [32]. Since polymorphisms for ACE2 and TMPRSS2 exist in different animal species [36,37,38,39,40], the analysis of the alignments of protein sequences such as ACE2 and TMPRSS2 can provide an estimate of the animal species potentially susceptible to SARS-CoV-2 [41,42,43]. Several recent studies [44,45] have suggested that due to the high conservation of ACE2, some animal species (i.e., cats, dogs, ferrets, tigers, and other wild species) are vulnerable to SARS-CoV-2 infection. A recent study which analyzed nucleotide sequences of 266 *ACE2* gene variants from 132 mammalian species showed that local similarities at key S protein-binding sites are good predictors of a high risk of mammals being infected by SARS-CoV-2 [46].

Similar to SARS-CoV-2 in humans, different variant strains in animal species were isolated, deriving mainly from human-to-animal transmission events in the latter stage of the pandemic. As reported by Cui et al. [33], based on the GISAID database [47], ferret, hippo, hyena, fishing cat, and binturong were only infected by the Delta variant; cat, dog, mink, deer, tiger, lion, snow leopard, gorilla, hamster, and otter were infected by more than one type of variant; and dog, cat, mink, deer, tiger, lion, snow leopard, and gorilla were also infected by non-variant strains.

In particular, cats and dogs, as the main animal species treated in this review, were infected by five (Alpha, Delta, Gamma, Lambda, and Omicron) and three (Alpha, Delta, and Omicron) variants, respectively [33].

#### 3.2.1. Cats

In Wuhan, a total of 102 cats were tested between January and March 2020. Fifteen (14.7%) were positive for RBD-based ELISA and eleven (10.8%) were positive via the virus neutralization test (VNT). Fifteen sera collected after the outbreak were positive for the receptor-binding domain (RBD) of SARS-CoV-2 via ELISA. Among them, 11 had SARS-CoV-2 neutralizing antibodies with a titer ranging from 1/20 to 1/1080. In addition, serum antibodies from two sampled cats reached the peak at 10 days after the first sampling and declined to the limit of detection within 110 days [48]. In France, in June 2020, one in twenty-two cats tested positive via an RT-qPCR rectal swab. This cat showed mild respiratory and digestive signs. A serological analysis confirmed the presence of antibodies against SARS-CoV-2 in two serum samples, taken 10 days apart. Genome sequence analysis revealed that the cat’s SARS-CoV-2 belonged to the phylogenetic clade A2a, like most of the French human SARS-CoV-2 [49]. Furthermore, in a study conducted in the United States, SARS-CoV-2 RNA was detected in two cats, 7–8 days after the onset of symptoms in the COVID-19 cohabiting human case [50]. Additionally, in Hong Kong, China, 50 cats were sampled at a time from the cohabiting owner’s onset of COVID-19 symptoms ranging from 3 to 15 (median: 8) days. SARS-CoV-2 RNA was detected in samples of 6 (12%) in 50 cats. Five of the positive cats came from confirmed families with COVID-19 infection, and only one cat belonged to an infected unconfirmed close contact. No infected cat developed signs of disease [51]. In Italy, neutralizing antibodies were detected in 1 out of 22 cats (4.5%) from COVID-19 positive families and in 1 out of 38 cats (2.6%) from households who tested negative for COVID-19. None of the sampled animals showed respiratory signs at the time of sampling [52,53]. Another cat with immunosuppressive conditions due to intestinal lymphoma developed signs of respiratory tract disease. The cat tested positive for SARS-CoV-2 viral RNA through RT-qPCR. Furthermore, the serological testing substantiated the presence of a SARS-CoV-2 infection with the detection of anti-SARS-CoV-2 antibodies. These data strengthen the assumption that comorbidities may play a role in the development of clinical disease [54]. The spread of SARS-CoV-2 was also studied in France in nine cats owned by veterinary students (*n* = 18), two of which tested positive for SARS-CoV-2. No animals obtained a positive result via RT-PCR of nasal or rectal swabs or for the presence of specific effects of SARS-CoV-2 [55]. Nevertheless, in Spain, an asymptomatic cat was reported for having viral antigens after cohabiting with a COVID-19 patient [56]. In Germany, the first large-scale survey of antibody occurrence in the domestic cat population was conducted. A total of 920 serum samples, collected from April to September 2020, were screened by an indirect multispecies ELISA. Overall, 0.69% (6/920) of serum samples were found to be positive for antibodies against SARS-CoV-2 via ELISA and iIFT [57]. In Israel, Kleinerman et al. [58] performed serological and molecular screening for SARS-CoV-2 in 131 cats in military bases, validating a novel quantitative serological microarray for use in cats that enables the simultaneous detection of IgG and IgM responses. Three of all of the analyzed cats showed IgG antibodies against SARS-CoV-2 RBD and S2P (2.3%), but none of the cats were positive for SARS-CoV-2 RNA via RT-PCR.

From April 2020 to October 2021, in Naples, 313 cats were tested for SARS-CoV-2, using nasopharyngeal, rectal swabs, and sera. Positive sera were from five cats (1.75%). The background revealed that four cats lived with COVID-positive owners, and three of which were symptomatic. Particularly, a 2-year-old female showed gastroenteric symptoms, a 15-year-old male showed mild lethargy and loss of appetite, and a 1-year-old female showed severe respiratory symptoms and died due to respiratory distress. Positive sera tested with ELISA were subjected to serum neutralization. They were found to be positive in the three symptomatic owned cats, with titers of 1:60, 1:80, and 1:160 [59].

Lenz et al. [60] detected and sequenced a SARS-CoV-2 Delta variant (AY.3) in fecal samples from an 11-year-old domestic cat with an owner who was positive for SARS-CoV-2. Sequencing of the feline-derived viral genomes from two fecal samples collected 7 days apart showed the two to be identical and differing by between 4 and 14 single nucleotide polymorphisms in pairwise comparisons to human-derived lineage “AY.3” sequences. These results confirm the repeated spillover infections of emerging SARS-CoV-2 variants that threaten human and animal health.

The susceptibility of cats to SARS-CoV-2 infection has been supported by several experimental observations [61,62,63,64,65]. Specifically, it has been shown that cats exposed to SARS-CoV-2 under laboratory conditions can become infected and are able to transmit the disease to other felines [66,67]. Laboratory experimental studies by Shi et al. [61] and Bosco-Lauth et al. [62] showed respiratory symptoms similar to those seen in humans, as well as injuries in the epithelia of the tracheal mucosa as well as the nasal passages and lungs. These observations clearly suggest a high relative susceptibility in the feline family with human-to-feline transmission recorded in domestic cats [61,63,66,68,69]. The similar susceptibility of cats to SARS-CoV-2 can also be due to this species sharing the angiotensin-converting enzyme 2 (ACE2) of the virus receptor with humans. Particularly, cat ACE2 is very similar in its structure to the SARS-CoV-2 spike-contacting regions of ACE2, with just two amino acids differentiating them. This could explain the mechanism of human-to-animal transmission and vice versa [36,70]. It is also necessary to emphasize that among the sequence analysis of ACE2s in other mammals, the most closely related sequence is from the domestic cat with an overall similarity of 85.2% in comparison to human ACE2 [36,70,71,72,73,74].

#### 3.2.2. Dogs

There have been several instances of SARS-CoV-2 infection in domestic dogs associated with presumed transmission from humans [53,75]. Two domestic dogs, positive via PCR test for SARS-CoV-2, have been reported in Hong Kong. Antibody responses were detected in both dogs using plaque reduction neutralization via oronasopharyngeal swabs. The viral genetic sequences of the dogs were identical to the virus detected in their respective human caregivers. Both Hong Kong dogs remained asymptomatic throughout the quarantine period [75]. A larger study of 603 dogs at the University of Bari in Italy revealed that only 15 dogs (3.3%) showed neutralizing antibodies to SARS-CoV-2, and none of the dogs showed clinical signs of disease at the time of sampling. Specifically, in samples from families with known COVID-19 status, neutralizing antibodies were detected in 6 out of 47 dogs (12.8%) and 1 out of 7 dogs (14.3) from families positive for COVID-19, and in 2 out of 133 dogs (1.5%) from COVID-19 negative households [53]. A significantly higher potential probability to test positive for neutralizing antibodies can be therefore inferred if the dog comes from a family known to be positive for COVID-19. Similarly, in a study in France, a remarkably high 21.3% (10 of 47 animals tested) of pets in COVID-19+ households tested positive, including 15.4% of dogs (2/13), highlighting a risk of seropositivity eight times higher for pets sharing a home with a COVID-19+ person than for pets in homes of unknown status [76]. A cross-sectional investigation for SARS-CoV-2 was conducted in Bangkok, Thailand, from June to September 2021. Out of a sample of 105 dogs, only 1 dog tested positive for SARS-CoV-2 and belonged to a positive COVID-19 family. Additionally, genome-wide sequence analysis identified the Delta variant SARS-CoV-2 [77]. In Rio de Janeiro, 9 in 29 dogs (31%) tested positive through nasopharyngeal samples taken from 11 to 51 days after the human index COVID-19 case onset of symptoms, developing non-relevant clinical signs and mild respiratory and gastrointestinal manifestations, with no associated laboratory abnormalities [78]. In Colombia, Rivero et al. [79] described the first event of symptomatic transmission in Latin America from a human to a dog by the B.1.625 lineage of SARS-CoV-2, finding 21 shared mutations in the complete genomes of viral sequences from owners and dogs. Therefore, the authors suggest that close contact between SARS-CoV-2-infected humans and pets should be avoided or limited to prevent the eventual emergence of novel mutations also dangerous to public health.

Laboratory studies suggest limited canine susceptibility to SARS-CoV-2; direct contact of healthy dogs with experimentally infected dogs did not cause viral spread, although neutralizing antibodies were detected 14 days after inoculation; furthermore, experimentally infected dogs never showed any clinical symptoms [61,62]. Several in vitro studies focus on the analysis of the amino acid sequence of the ACE2 receptor, which in dogs is 81% identical to humans’ [32,40]. However, dogs possess only one of the four amino acid sequences of the ACE2 receptor, whose functions have been closely linked to increased susceptibility to SARS-CoV-2 [36]. Furthermore, a recent study suggests that domestic dogs lack some genes responsible for an inflammatory reaction that occurs in humans after contagion. Therefore, it is assumed that this is one of the plausible reasons for the little overt clinical symptoms in the dog [80].

#### 3.2.3. Other Animal Species

SARS-CoV-2 infection was detected in mink farms in the Netherlands starting from April 2020 [81,82], and was subsequently found in other countries, i.e., Spain, Utah (USA), Italy, Sweden, Greece, France, Poland, and Canada [81]. Infected animals developed respiratory diseases with pathological findings typical of viral pneumonia [83]. Several scientists found close correlations between viral sequences obtained from vison samples with those of human origin, suggesting a potential likelihood of transmission [84,85]. Furthermore, the proven transmission between mink housed separately suggests diffusion via respiratory droplets or aerosols, which is consistent with the detection of viral RNA in the dust of infected environments [81,86]. SARS-CoV-2 has been detected in four tigers and three lions at a zoo in the Bronx, New York [87]. Epidemiological and genomic data indicated human-to-tiger transmission, supporting a close evolutionary relationship between viral strains in tigers and tiger guardians. On the other hand, no clear source of transmission was identified for the lions, as the genomic sequences were divergent. The animals developed mild respiratory symptoms. Subsequently, an exotic puma (July 2020) and three African lions (July 2021) in the same zoo in Johannesburg, South Africa, tested positive after contact with an infected handler [88]. The characteristics of ACE2 predict a high susceptibility to SARS-CoV-2 among primates [41,89], and this is confirmed by the results of experimental studies [90,91] and by the naturally acquired infection in captive gorilla [92]. Furthermore, at the San Diego Zoo Safari Park, a group of eight gorillas tested positive for SARS-CoV-2. The animals presented mild symptoms such as coughs and airway congestion. Similarly, the susceptibility of cervids is based on the binding affinities of ACE2 receptors [41]. Hale et al. [93] found SARS-CoV-2 in 129 of 360 (35.8%) white-tailed deer from northeastern Ohio tested via rRT-PCR between January and March 2021, documenting that this species is highly susceptible to SARS-CoV-2 infection. This makes white-tailed deer able to sustain transmission in the wild, potentially opening new evolutionary pathways such as transmission to other wild species.

Experimental infections also resulted in evident subclinical infections and elimination of the virus via nasal secretions and feces. It is important to emphasize that indirect contact animals were infected and spread infectious viruses, indicating efficient SARS-CoV-2 transmission from inoculated animals [94]. Ulrich et al. [95] reported that cattle had a low susceptibility to experimental SARS-CoV-2 infection, while experimental models of pigs, chickens, ducks, quail, and geese have shown a lack of susceptibility to the virus [61,96,97]. Furthermore, ferrets were found to be highly susceptible to SARS-CoV-2. To study the replication dynamics, groups of three ferrets were experimentally inoculated. The virus was detected in nasal washes of all tested animals [61], and SARS-CoV-2 was efficiently transmitted to three ferrets via direct contact, but none of the animals had any obvious clinical signs [96].

### 3.3. Infections in Humans

The COVID-19 disease first outbreaks were perceived as clusters of pneumonia cases in Wuhan, China. Later on, the identification and confirmation of a novel coronavirus disease allowed medical physicians to pinpoint other symptoms associated with the disease. Studying 41 COVID-19 patients admitted to a Wuhan hospital, Huang et.al reported that in addition to the bilateral ground glass opacity observed in all patients, they also presented fever (98%), cough (76%), dyspnea (55%), and fatigue (44%). Other symptoms such as sputum production (28%), headache (8%), hemoptysis (5%), and diarrhea (3%) were also observed. Severe symptomatology included acute respiratory distress syndrome (29%), acute cardiac injury (12%), acute kidney injury (7%), and shock (7%) [98]. These symptoms were later observed worldwide as the pandemic progressed, adding on other symptoms such as loss or change in smell and taste, loss of appetite, dizziness, chills, runny nose, abdominal pain, and vomiting [99,100,101].

Although fatigue and pain seem to be the most prevalent long-COVID symptoms, to date the association of SARS-CoV-2 with COVID symptoms is still unclear and this problem can be explained by the different variants [102]. In 2020, four main variants were considered by WHO among the VOCs (variants of concern): (1) Alpha, identified for the first time in the UK in September 2020; (2) Beta, identified for the first time in South Africa in May 2020; (3) Gamma, first identified in Brazil in November 2020; and (4) Delta, first identified in October 2020 in India [103]. Interestingly, 2021 and 2022 were characterized by the Omicron variant, detected in South Africa in November 2021, which rapidly replaced Delta as the main circulating variant [104]. Scientists described Omicron as the worst variant to emerge since the start of the pandemic, with 32 mutations in the spike protein, about double the number of mutations presented by the Delta variant [105]. Currently, there are about six variants that concern WHO, which are monitored daily and represent 72.9% of the prevalence of infections. These include BQ.1-Cerberus (42.5%), Omicron BA.5, being one of the most among five mutations (13.4%), BA.2.75-Centaurus (9.8%), the recombinant XBB Gryphon (6.1%), BA.4.6 (1%), and BA.2.30.2 (0, 1%).

There are several COVID-19 vaccines which have been validated for use by WHO. The first mass vaccination program started in early December 2020 and the number of vaccination doses administered is updated on a daily basis on the COVID-19 dashboard. Actually, there are four categories of vaccines in clinical trials: whole virus, protein subunit, viral vector, and nucleic acid. Although these vaccines have saved millions of human lives, they have not eliminated the virus [106]. Thus, as SARS-CoV-2 continues to evolve, it is necessary to better evaluate the timing and implementation of additional COVID-19 vaccine doses, to rapidly expand scientific knowledge on these variants, to track the spread and virulence of the virus, and to provide advice to countries and individuals on measures to protect health and prevent the spread of new outbreaks.

## 4. Serological and Molecular Diagnosis

The worldwide rapid spread of the COVID-19 disease has created the need to find rapid and accurate diagnostic methods for effective clinical and public health management. The scientific community has focused on finding optimal diagnosis methods in order to ensure the rapid treatment and or isolation of infected individuals.

The main COVID-19 diagnosis methods fall into nucleic acid amplification to detect the virus genome, serological testing to detect antibodies raised against viral antigens, and antigenic testing to detect viral antigens. Auxiliary methods include symptomatology and lung imaging [107,108,109].

The current review aims to discuss the most applied diagnosis methods, elucidating their specific properties, strengths, and weaknesses.

### 4.1. SARS-CoV-2 Testing in Humans

The most performed nasopharyngeal tests so far have been real-time reverse-transcriptase polymerase chain reaction RT-PCR and the antigenic test. In January 2020, a few months into the pandemic, the genetic material of the virus was sequenced, and important genes were discovered [110]. Primers and probes were designed to develop the RT-PCR diagnosis test, which is still the gold standard method for COVID-19 diagnosis [110,111]. RT-PCR detection presents many advantages as it is rapid, precise, highly sensitive, and specific. However, due to its dependence on factors such as viral load, RNA sequence variation, sampling, and proper sample handling, it is highly subjected to false negatives.

In an effort to overcome RT-PCR’s downsides, alternatives such as digital PCR and RT-LAMP have been in development [112,113,114].

RT-PCR is still widely used for COVID-19 diagnosis despite its drawbacks. Underway research to find better techniques remains infructuous due to the lack of personnel with the required experience and interpretation skills [109].

COVID-19 antigen tests exploit the antigen–antibody reaction to detect the presence of the SARS-CoV-2 spike and nucleocapsid protein in oropharyngeal or nasopharyngeal swabs of infected individuals. Enzyme-linked immunosorbent assay (ELISA), enzyme immunoassay (EIA), chemiluminescence immunoassay (CLIA), and lateral flow immunoassay (LFIA) are examples of techniques employed for antigen testing. With the high demand for testing, rapid antigenic tests, mainly based in LFIA with visual readout, became dominant in the market [107,110]. These tests have the advantage of being portable, easy, and economic but also pose a problem regarding the accuracy and the concordance between different tests. A pool of 58 antigen test studies performed confirmed that positive individuals resulted in average sensitivities of 72% and 58% for symptomatic and asymptomatic people, respectively.

According to the literature, neutralizing antibodies against SARS-CoV-2 were found in half of the infected individuals by Day 7 and in all individuals by Day 14 [108]. That said, up to 300 serological tests have been developed with the aim of detecting antibodies raised against viral antigens, namely the spike protein and the nucleocapsid protein. These tests are of minimal interest for the POCT—point of care test—but have been crucial for community studies, the detection of past infections, and even to test vaccine effectiveness [107,115]. Similarly to antigenic tests, serological tests are based on techniques such as ELISA and chemiluminescence immunoassay (CLIA), and rapid diagnostic tests are mainly based on LFIA—lateral flow immunoassays. All methods employ recombinant antigens to detect the presence of neutralizing agents such as IgM, IgG, and IgA [107,116].

Symptomatology and lung imaging have proven useful as auxiliary diagnosis methods of COVID-19 disease. Indeed, a study conducted on one million people in England infers that some symptoms such as fever, loss or change in sense of smell or taste, persistent cough, chills, appetite loss, and muscle aches jointly are a good prediction of a COVID-19 positive test [100]. COVID-19-related pneumonia abnormalities can be observed through computed tomography or lung ultrasound, which has the advantage of being radiation-free. Similarly to symptomatology, these imaging techniques are a good predictive method but are also useful for locating the infection [114]. Both symptomatology and imaging were very important during the first period of the pandemic, when testing was scarce, but have not proven to have relevant accuracy as they are uncertain. Furthermore, COVID-19 symptoms and pneumonia lesions are not exclusive to the disease; thus, these techniques cannot be reliably used on their own [109,110].

After more than three years into the pandemic, testing has proven to be key to the infection’s control.

However, although laboratory tests are simple, accessible, and low-cost, methods to evaluate SARS-CoV-2 infection, their accuracy, and specificity remain controversial [117].

### 4.2. SARS-CoV-2 Testing in Animals

SARS-CoV-2 tests on animals, similar to those used on humans, involve both the detection of active infection and previous exposure. Indeed, as reported by the literature consulted, to detect active infection, molecular polymerase chain reaction (PCR) tests, virus isolation, and nucleoprotein (N) antigen tests are performed [118,119]. In particular, real-time reverse transcription PCR is the most commonly used molecular assay in the detection of SARS-CoV-2 in cats [60,120], dogs [75], and large cats [88,118,119,121].

Unlike PCR which shows greater sensitivity, antigen testing has been shown to have greater specificity and produce fewer false positives [122]. In addition to these routine testing methods, sequencing, including next-generation sequencing, is usually conducted to characterize strains involved in outbreaks [118,119,121].

For the detection of previous exposure to SARS-CoV-2 by assessing antibody immune responses, the virus neutralization test (VNT), the surrogate virus neutralization test (sVNT), and the enzyme immunoadsorption test were used (ELISA). Unlike conventional VNT which also requires laboratory biosafety level 3 (BSL3), sVNT uses the interaction of the SARS-CoV-2 receptor-binding domain and ACE2, which will be blocked by specific viral antibodies in serum samples [123]. Therefore, the sVNT assay skips the stringent requirement of BSL3 and can be applied to different animal species.

ELISA has also been used to determine antibody responses in animals [124,125]. For the detection of antibodies against nucleoprotein (N), a commercial double-antigen polyspecific ELISA was used for all sensitive animal species [124]. An in-house developed species-specific ELISA [125] and a new quantitative serological microarray [58] were also performed for the detection of SARS-CoV-2 antibodies.

## 5. Potential Transmission Risk by SARS-CoV-2 in Human–Animal Relationship

Humans, animals, and the environment play significant roles in the emergence and transmission of various infectious diseases. According to the World Health Organization (WHO), any disease or infection that is naturally transmissible from vertebrate animals to humans or from humans to animals is classified as a zoonosis (WHO) [126,127]. Anthropogenic changes in the ecosystem have resulted in an increase in shared habitats between humans and animals, offering multiple pathways of translocation for the spread of emerging infectious diseases [128]. The current literature focuses on the emerging pathogens of animal origin [5,126,129], while scientific papers rarely mention the role of humans in the variety of emerging diseases affecting the animal kingdom [130]. Currently, both animals and humans suffer from the negative effects of a changed human–animal bond; therefore, the concept of bidirectional zoonosis, that is, the transmission of high-risk and multi-resistant pathogens from humans to animals, should not be underestimated [131,132]. Indeed, a growing number of reports indicates that human pathogens can colonize and infect companion animals, thus becoming further widespread in the home environment [133,134,135]. Nowadays, 50% of owners allow their pets to lick their face; 60% of animals visit the bedroom; and 30% sleep with the owner in bed [136]. The recent trend towards closer contact between humans and animals is responsible for the increased risk of zoonotic infections through biting, licking, scratching, sneezing, or coughing, sharing the same living environment or body fluids or secretions, and indirect contact through contaminated bedding, food, water, or bites of an arthropod vector [137]. In the last two decades, the occurrence of three outbreaks of CoV with zoonotic origin has highlighted the epidemiological role of animals in a public health context. In these terms, the current SARS-CoV-2 epidemic might represent a concrete example of the consequence of the mutated human–animal and environment relationships.

Current scientific evidence demonstrates a proven risk of exposure and infection for animals with SARS-CoV-2, suggesting that exposure may result in asymptomatic/paucisymptomatic infections in animals. Now, there is no strong scientific evidence that animals played a fundamental role in the epidemic spread of SARS-CoV-2. Indeed, human-to-human contagion is considered the main transmission route. Despite this likelihood, to date, animals have not been adequately considered a worrying source of concern for people [32,57,58]. Dogs and cats are among the most popular companion animal species, often living in close contact with their owners. Therefore, in this context, risk analysis assumes an important epidemiological value. The risk of infection is, indeed, closely related to the state of health of the people with which the animal comes into contact and to the context in which the risk assessment is carried out (Figure 2). Immunosuppressed people, such as elderly patients, pregnant women, children, or patients with comorbidities, are at greater risk of contracting infections caused by zoonotic pathogens, and the consequences of which are often much more severe than in immunocompetent individuals and can occasionally be fatal [138,139,140,141]. In these terms, veterinarians play a key role in disclosing clear information aimed at preventing transmission risks. The recommendations concern adequate hygienic practices, nutrition, arrangement at home, and responsible breeding, which are essential to prevent the negative implications of human–animal bonds [137]. Furthermore, appropriate risk planning in relation to the different users and contexts in which animals are placed would allow for the implementation of targeted health guidelines and protocols. Therefore, addressing the issue of transmission risk and approaching a realistic assessment of the risk of infections requires collaboration between veterinarians, physicians, public health professionals, and epidemiologists, as proposed by the One Health Initiative (One Health Initiative).

## 6. Discussion

Infectious diseases have always constituted an enormous threat to humanity, particularly in large communities where they have caused the downfall of entire civilizations. Among them, zoonoses have raised questions regarding the risks of animal and human cross-contamination, particularly in the context of animal breeding and domestication.

Actually, there are many zoonotic diseases, some with a reduced severity index and others being decidedly serious for animal and human health. Thus, correct and effective prevention is necessary to reduce the risk of wide spreading. In this regard, some general hygiene rules are of particular importance which must always be respected and which concern the following: (1) hygiene and control of the good state of health of the animal; (2) hygiene of personal contact with the animal; (3) animal feed hygiene; (4) hygiene of animal droppings; and (5) fight against stray animals.

However, faced with the risk of serious epidemics, international health authorities cannot only rely upon existing surveillance, warning, and response systems, which could be inadequate and unequipped to deal with unforeseen situations. Several elements contribute to the progressive and growing emergence of pathogens; among these, we consider the amplified rates of global diffusion, the climatic and anthropogenic changes in the ecosystem which extend the amount of habitat shared between humans and animals, the growing close contact with animals, and, above all, the close relationship that develops between humans and domestic animals. From this point of view, the SARS-CoV-2 pandemic becomes a powerful demonstration of the close links between human health and domestic, wild, and farmed animals [67,85]. Actually, the literature at our disposal highlights the proven risk of exposure and infection of animals to SARS-CoV-2, and, although it is still not clear whether they could be a worrying source of infection for humans, the animal species closest to humans have always represented a potential vector for the transmission of numerous zoonotic agents [142]. Therefore, close proximity to humans and the social interactions that are established offer continuous opportunities for the transmission of pathogens between different species. Another important issue which has emerged during the COVID crisis has been the great diversity in coronaviruses circulating in humans and in domestic and wild animals. To fight COVID-19 and, in general, all of the zoonoses in a more integrated and global way, new response mechanisms would be needed through the use of appropriate tools including new technologies, molecular biology, and analytical epidemiology. Furthermore, an international collaboration between different disciplines such as medicine, veterinary medicine, biology, and bioinformatics should be improved to achieve these goals, recognizing that human and animal health and the ecosystem are inextricably linked, following the “One Health” concept [143].

## Figures and Tables

**Figure 1 microorganisms-11-00514-f001:**
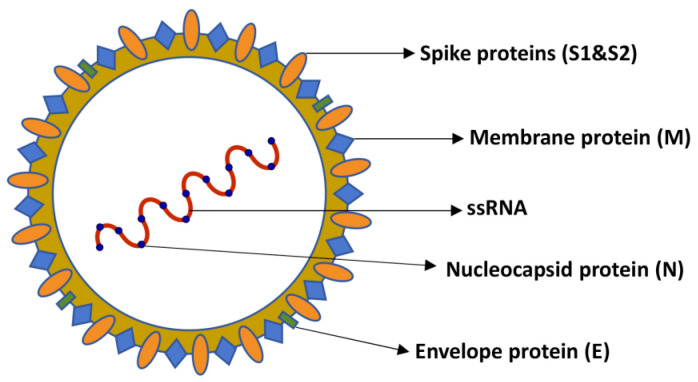
Schematic representation of SARS-CoV-2: the structure of SARS-CoV-2 is formed by the structural proteins, including the spike (S), membrane (M), envelope (E), and nucleocapsid (N) proteins that are involved in morphogenesis, viral assembly, host infection, and membrane fusion.

**Figure 2 microorganisms-11-00514-f002:**
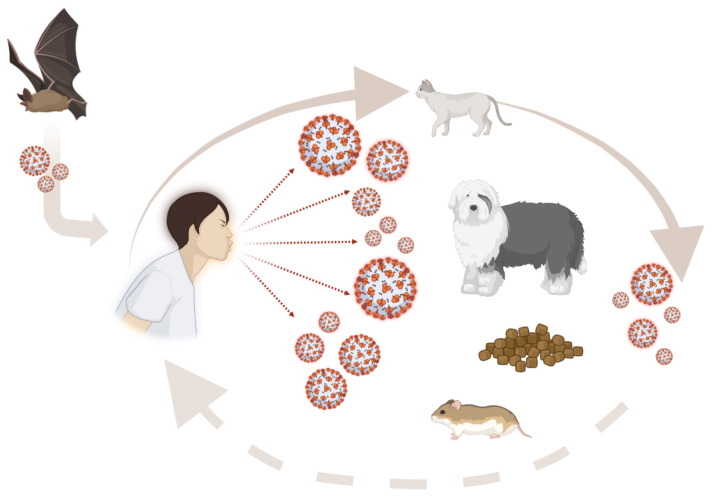
Possible mode of transmission of SARS-CoV-2 in humans and animals: Animal species closest to humans have always represented a potential vector for the transmission of numerous zoonotic agents. However, it is still not clear whether they could be a worrying source of infection for humans. This figure has been created using BioRender.com.

## Data Availability

Not applicable.

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
