# Peer review of "SARS-CoV-2 Affects Both Humans and Animals: What Is the Potential Transmission Risk? A Literature Review"

_microorganisms, 2023, doi:10.3390/microorganisms11020514_

Round 1
Reviewer 1 Report
Dear authors,
Thank you for the review submitted to Microorganisms.
I have many concerns about the currently provided text.
Two major ones:
1) The title of the review is inviting to read about the potential risk of transmission. In my opininon, much more detail should be given to ACE-2 structure and variability in humans and animal hosts as well as details of RBD in SARS-CoV-2 should be presented in connection with different (at least major) knows variants. It should be discussed how different mutations impact receptor-binding and increase or decrease the potential risk of transmission in humans and animals. In my opinion, currently there is no or too little information in the review on this.
2) You submitted the text in Autumn 2022, while the literature search was performed until April 2021. That is very out of date in my view as it seems no problem at all to add the whole 2021 and at least 9 months of 2022 to this review.
Also, I suggest that any big review should have illustrations (figures or schemes) for better and easier reading. Please consider adding them.
The other points to address are:
- Introduction: too long and too many historical details that have nothing to do with transmission risks. Interestingly, SARS-CoV-2 is mentioned once in the last paragraph. Seems strange.
- Main section:
o Please take care how you write SARS-CoV-2. It should be uniform, and you write it as “SARS-CoV2, Sars-CoV-2, SARS-Cov-2, etc.”
o 3.1 – is very general and is not needed in 2022. You miss the N-protein in description of structural proteins. Lines 143-144 – that is the genome of CoVs, not SARS-CoV-2, it is quite stable in its length.
o 3.2 – 3.3 – I suggest adding much more detail on molecular biology of the virus and the receptor in order to explain the transmission and its possible risk. Currently, these sections are too general.
o 3.3 – 4 - I suggest removing it completely. The information about humans is very outdated, and currently many if not all figures and estimates have changed with different variants of the Omicron circulating around the world.
o Section 5 about the bi-directional transmission risk is too general and looks more like the Discussion, please, provide the exact details and examples related to the bidirectional transmission of SARS-CoV-2
o Section 5.1. - is mainly focused on the type of different AAI rather than on the risk of transmission. Consider revising.
- Discussion: has too many repeats from the introduction, and is not discussing the transmission risks but more general aspects of COVID-19, symptoms, pets, AAIs and a bit about how COVID-19 could migrate to animals and back. Too general for a focused discussion about transmission risks.
In general I would conclude that the review should be completely rewritten with more detail on what is necessary for transmission from the virus side and from the host, giving more details for cats, dogs, ferrets and other animals, discuss potential independent evolution of SARS-CoV-2 in wild nature and avoid any description of testing methods and description of SARS-CoV-2 in humans.
Reviewer 2 Report
The title indicates a relevant question in the field of SARS-CoV-2 research/diagnostics/epidemiology, i.e. the transmission risk between humans and animals.
Important points that should be thoroughly addressed are:
1. Direction of transmission: Is the transmission primarily from humans to animals or from animals to humans or does it occur equally in both directions?
2. Animal species with reported infection and symptoms: Which animal species are known to be infected? Which symptoms do the animals show and how does this compare to humans?
3. Diagnostics: Which diagnostics that are used for humans can be used for animals? Is there a need to further develop diagnostics for animals?
4. Evolution of SARS-CoV-2 in animals: (How) does SARS-CoV-2 evolve in animals?
However, the review does not satisfactorily address these points.
Major points of criticism:
1. Section 1: The authors introduce the terms “pandemic” and “zoonosis”. It might be of interest to review here major zoonotic diseases that are caused by viruses. Instead, the authors give a brief overview of past pandemics which is not relevant to the main question of the review.
2. Section 3.2: Subsections should read 3.2.1, 3.2.2, and 3.2.3.
3. Section 4. Serological and Molecular Diagnosis: In this section the authors focus on the diagnostics used for humans. They should instead focus on the question which diagnostics that are used for humans can be used for animals and if there is a need to further develop diagnostics for animals. Table 1 is thus irrelevant.
4. Section 5:
- The important information is given in ll. 483 to 488. This section should in more details and more clearly address the question of the direction of transmission.
- This section might be combined with section 2.
- Section 5.1. is irrelevant to the main question of the review.
5. Section 6: This section is superfluous/redundant.
Round 2
Reviewer 1 Report
Dear authors,
Thank you for updating the review. However, some of the issues remain and need to be corrected. First of all, you write about the 15 species that were infected with SARS-CoV-2. But even Cui et al., that you mention in your review have 18 in their text, while the FAO website has much more (more then 25, please, have a look https://www.fao.org/animal-health/situation-updates/sars-cov-2-in-animals/en). It seems reasonable to update this and also provide a few words about this link for the professionals and other readers. I also believe that some of the aspects are still not discussed - i.e. that the virus can have its own evolution trend in the wild life once being transmitted, and a good example of this is the deer described by the american team. You only briefly mention the possibility of the deer to be infected, however, it seems important to give an overview of how this transmission toute can influenza the virus evolution in nature (outside the human population). The very brief information about the receptor-binding properties of different SARS-CoV-2 has been given but you miss the important paper by Wei et al (https://doi.org/10.1038/s41598-020-80573-x), I would advise to add a few lines about this as well.
I still believe the section 4.1. is too long and the newly added 4.2 could be of more detial given the context of the review.
Reviewer 2 Report
As mentioned in the first review report, a main shortcoming of the review is that is does not satisfactorily address the question asked in the title, i.e. “what is the potential transmission risk?”. The revised version of the manuscript still does not adequately address this major point of criticism.
Response to authors’ responses:
1. Direction of transmission: Is the transmission primarily from humans to animals or from animals to humans or does it occur equally in both directions?
Re: We thank the Reviewer for the suggestion. As shown in figure number 2, SARS-CoV-2 is of animal origin, adapted to humans and for the reasons outlined in the sections dedicated to animal species, it can infect the latter which could become an occasional source of a new infection.
Re: The authors only refer to Figure 2. The authors should instead have expanded section 5., referencing studies that have provided evidence for transmission from humans to animals or vice versa. This part could also be integrated into subsection 3.2.
2. Animal species with reported infection and symptoms: Which animal species are known to be infected? Which symptoms do the animals show and how does this compare to humans?
Re: We have integrated the information highlighted by the Reviewer related to animal species in subsection 3.2 and in subsubsections 3.2.1, 3.2.2, 3.2.3.
Re: The authors do not mention at any point the evidence supporting the zoonotic origin of SARS-CoV-2 (Wuhan wet markets). Furthermore, the question how the severity of the symptoms observed in animals compares to those in humans (Is the course of disease in general milder? Is the mortality rate lower?) has not been answered directly by the authors. Another open question the authors could have addressed is whether there are animal species known to be resistant to infection with SARS-CoV-2.
3. Diagnostics: Which diagnostics that are used for humans can be used for animals? Is there a need to further develop diagnostics for animals?
Re: We thank the Reviewer for the suggestion. We have integrated the lacking information about diagnostic tools for animals. Thank you.
Re: Overall, this point has been satisfactorily addressed by the authors.
4. Evolution of SARS-CoV-2 in animals: (How) does SARS-CoV-2 evolve in animals?
Re: At present we can only report what has been reported in the literature with respect to the description of the clinical pictures in infected animals and we can state that a potential risk of transmission from animals to humans cannot be excluded. In this regard, we have tried to improve the "conclusions" section making it more consistent with the entire paper. Thanks for the valuable suggestion.
Re: Accepted
1. Section 1: The authors introduce the terms “pandemic” and “zoonosis”. It might be of interest to review here major zoonotic diseases that are caused by viruses. Instead, the authors give a brief overview of past pandemics which is not relevant to the main question of the review.
Re: We thank the Reviewer for raising this interesting point. As suggested, the length of the introduction has been reduced, focusing on the main zoonotic diseases caused by viruses.
Re: The authors still give an overview of the past pandemics, not a review of the major zoonotic diseases.
2. Section 3.2: Subsections should read 3.2.1, 3.2.2, and 3.2.3.
Re This error has been corrected and the number of the paragraphs have been changed.
Re: Accepted
3. Section 4. Serological and Molecular Diagnosis: In this section the authors focus on the diagnostics used for humans. They should instead focus on the question which diagnostics that are used for humans can be used for animals and if there is a need to further develop diagnostics for animals. Table 1 is thus irrelevant.
Re: As suggested by the Reviewer we added the diagnostic methods also in animals and we removed the Table 1.
Re: Accepted
4. Section 5:
- The important information is given in ll. 483 to 488. This section should in more details and more clearly address the question of the direction of transmission.
- This section might be combined with section 2.
- Section 5.1. is irrelevant to the main question of the review.
Re: As requested by the Reviewer, we have rewritten the discussion focusing on the transmission risks and including the paragraph about the AAI (5.1).
Re: The authors probably mean that section 5. (not the discussion) has been revised and that subsection 5.1 has been removed (not included). Subsection 5.1 has not really been removed, but integrated into section 5. Thus, this section still focuses too much on AAIs, whereas the focus of this section should be the transmission risk form humans to animals and vice versa.
5. Section 6: This section is superfluous/-redundant.
Re: Please, see above.
Re: Section 6 has not been changed. Thus, this point of criticism has not been addressed.
Other points of criticism:
1. Ll. 86/87 “The purpose of this review is to provide a comprehensive analysis about the impact of coronavirus disease 2019 (COVID-19) on humans and domestic animals.”: This does not match the title which indicates that the focus of the review is the transmission risk.
2. L. 172: What does “non-variant strains” mean? Every genetic lineage that is not the parental, wildtype strain is a variant; variants are further subdivided into VOCs, VOIs, and VUMs (as well as subvariants).
3. Ll. 211/212 “performed a serological and molecular genetic screening for SARS-CoV-2”: What do the authors mean by “genetic”? In this study serological analyses and PCR assays were performed.
4. New/revised sentences and paragraphs: Extensive editing of English language and style are required.

Round 3
Reviewer 1 Report
Dear authors,
Thank you for the efforts, now the paper has become much better and logical. Please consider small revision:
line 100 - you have the link (Cui Nature 2019) directly instead of [#]
line 163 - "in tigers" should be just "tigers"
line 175 "SARS-Cov-2" should be "SARS-CoV-2
line 380 - "covid-19" should be "COVID-19"
lines 424, 425, 430 "Covid-19" should be "COVID-19"
Reviewer 2 Report
Response to authors’ responses:
1. Direction of transmission: Is the transmission primarily from humans to animals or from animals to humans or does it occur equally in both directions?
Re: The authors only refer to Figure 2. The authors should instead have expanded section 5., referencing studies that have provided evidence for transmission from humans to animals or vice versa. This part could also be integrated into subsection 3.2.
Re: We apologize for not having adequately supplemented section 5 as indicated in the previous revision. As suggested, we have modified and deepened both subsection 3.2.
Re: The new text in subsection 3.2 and section 5 requires editing:
- Subsection 3.2:
o Ll. 148-150: => “SARS-CoV-2 infection in animals is usually asymptomatic but can occasionally cause anything fromsymptoms ranging from mild respiratory and gastrointestinal symptoms to pneumonia and death [30].”
o Ll. 166-171: The new sentences should be better integrated in this paragraph:
“Several recent studies [41, 42] have suggested that due to the high conservation of ACE2, some animal species (i.e., cats, dogs, and ferrets, in tigers and other wild species) are vulnerable to SARS-CoV-2 infection. A recent study which analyzed nucleotide sequences of 266 ACE2 gene variants from 132 mammalian species showed that local similarities at key S protein-binding sites are good predictors of a high risk of mammals of being infected by SARS-CoV-2 [48].”
o Ll. 172-174: => “Similar to SARS-CoV-2 in human, …”
o Ll. 174-175: “The last update published by FAO dating back to December 2022, mentions about 30 animal species naturally infected (RNA detection) by SARS-Cov-2.” => This sentence should be placed before the last sentence of the first paragraph of subsection 3.2.
- Section 5:
o Ll. 485-491: => “Current scientific evidence demonstrates a proven risk of exposure and infection toof animals to SARS-CoV-2, suggesting that exposure may result in asymptomatic/pauci-symptomatic infections in animals. Now, tThere is no strong scientific evidence that animals play a fundamental role in the epidemic spread of SARS-CoV-2; which instead recognizes human-to-human contagion is considered as the main transmission route. Despite this likelihood, to date animals are not adequately considered a worrying source of concern for people [32, 57, 58].”
2. Animal species with reported infection and symptoms: Which animal species are known to be infected? Which symptoms do the animals show and how does this compare to humans?
Re: The authors do not mention at any point the evidence supporting the zoonotic origin of SARS-CoV-2 (Wuhan wet markets). Furthermore, the question how the severity of the symptoms observed in animals compares to those in humans (Is the course of disease in general milder? Is the mortality rate lower?) has not been answered directly by the authors. Another open question the authors could have addressed is whether there are animal species known to be resistant to infection with SARS-CoV-2.
Re: We thank the reviewer for this suggestion. We have mentioned Huanan Seafood Wholesale Market in the introduction. We are very sorry and apologize for the lack of response but in subsections 3.2.1, 3.2.2., 3.2.3, dedicated to animal species susceptible to SARS-CoV-2 infection, as well as reporting epidemiological data and reports of cases, we had already indicated the relative symptoms, when reported (e.g., lines 188-190; 224-226). In addition, at lines 144-146 we have added in general how the symptomatology evolves in sensitive animals. Regarding the question on resistant animal species, we believe that it is probably too early to report this data and above all the scientific literature currently only shows positivity data.
Re: Accepted
1. Section 1: The authors introduce the terms “pandemic” and “zoonosis”. It might be of interest to review here major zoonotic diseases that are caused by viruses. Instead, the authors give a brief overview of past pandemics which is not relevant to the main question of the review.
Re: The authors still give an overview of the past pandemics, not a review of the major zoonotic diseases.
Re: We apologize for the omission in the requested correction. As required, we have rewritten the introduction focusing on the main zoonoses.
Re: The authors have now indeed focused on the main zoonoses, but could have described them in a more systematic way, giving briefly for each zoonosis information about:
- Symptoms
- Treatment (if available)
- Vaccination (if available)
- Number of infections/deaths
Other comments:
- L. 30: “when, man”
- L. 36: => “have been played”
- Ll. 40-56: Split up into two paragraphs (1. Black Death, 2. Rabies)
- L. 67: “AIDSthis infection”
- Ll. 75-85: Merge into one paragraph.
3. Section 5:
- The important information is given in ll. 483 to 488. This section should in more details and more clearly address the question of the direction of transmission.
- This section might be combined with section 2.
- Section 5.1. is irrelevant to the main question of the review.
Re: The authors probably mean that section 5. (not the discussion) has been revised and that subsection 5.1 has been removed (not included). Subsection 5.1 has not really been removed, but integrated into section 5. Thus, this section still focuses too much on AAIs, whereas the focus of this section should be the transmission risk form humans to animals and vice versa.
Re: We apologize for the omission in the requested correction. We have removed the too detailed description of AAIs, in favor of the potential risk of transmission.
Re: Accepted
5. Section 6: This section is superfluous/-redundant.
Re: Section 6 has not been changed. Thus, this point of criticism has not been addressed.
Re: We apologize for the omission in the requested correction. As required, we have rewritten the discussion and we have combined it with the conclusions.
Re: Accepted
Other points of criticism:
Re: Accepted
